# Characterization of T cell responses to co-administered hookworm vaccine candidates Na-GST-1 and Na-APR-1 in healthy adults in Gabon

Yoanne D. Mouwenda[1,2]*, Madeleine E. Betouke Ongwe[1,2,3☯], Friederike Sonnet[2☯], Koen A. Stam[2], Lucja A. Labuda[2], Sophie De Vries[1,4], Martin P. Grobusch[1,4,5], Frejus J. Zinsou[1,5], Yabo J. Honkpehedji[1,5], Jean-Claude Dejon Agobe[1,4,5], David J. Diemert[6], Remko van Leeuwen[7], Maria E. Bottazzi[8], Peter J. Hotez[8], Peter G. Kremsner[1,5,9], Jeffrey M. Bethony[6], Simon P. Jochems[2], Ayola A. Adegnika[1,2,5,9], Marguerite Massinga Loembe[1], Maria Yazdanbakhsh[2]

1 Centre de Recherches Médicales de Lambaréné (CERMEL), Lambaréné, Gabon, 2 Department of Parasitology, Leiden University Medical Center (LUMC), Leiden, The Netherlands, 3 Centre National de la Recherche Scientifique et Technologique (IRET- CENAREST), Libreville, Gabon, 4 Center of Tropical Medicine and Travel Medicine, Department of Infectious Diseases, Division of Internal Medicine, Amsterdam University Medical Center, (AMC), University of Amsterdam, Amsterdam, the Netherlands, 5 Institut für Tropenmedizin, Universität Tübingen, Tübingen, Germany, 6 Department of Microbiology, Immunology and Tropical Medicine, School of Medicine and Health Sciences, The George Washington University, Washington, District of Columbia, United States of America, 7 Amsterdam Institute for Global Development (AIGHD), Amsterdam, The Netherlands, 8 Texas Children's Hospital Center for Vaccine Development, Departments of Pediatrics and Molecular Virology and Microbiology, National School of Tropical Medicine, Baylor College of Medicine, Houston, Texas, United States of America, 9 German Center for Infection Research, Tübingen, Germany

☯ These authors contributed equally to this work.
* Y.D.Mouwenda@lumc.nl

**Data Availability Statement:** All relevant data are within the manuscript and its Supporting Information files.

## Abstract

Two hookworm vaccine candidates, Na-GST-1 and Na-APR-1, formulated with Glucopyranosyl Lipid A (GLA-AF) adjuvant, have been shown to be safe, well tolerated, and to induce antibody responses in a Phase 1 clinical trial (Clinicaltrials.gov NCT02126462) conducted in Gabon. Here, we characterized T cell responses in 24 Gabonese volunteers randomized to get vaccinated three times with Na-GST-1 and Na-APR-1 at doses of 30µg (n = 8) or 100µg (n = 10) and as control Hepatitis B (n = 6). Blood was collected pre- and post-vaccination on days 0, 28, and 180 as well as 2-weeks after each vaccine dose on days 14, 42, and 194 for PBMCs isolation. PBMCs were stimulated with recombinant Na-GST-1 or Na-APR-1, before (days 0, 28 and 180) and two weeks after (days 14, 42 and 194) each vaccination and used to characterize T cell responses by flow and mass cytometry. A significant increase in Na-GST-1 -specific CD4$^+$ T cells producing IL-2 and TNF, correlated with specific IgG antibody levels, after the third vaccination (day 194) was observed. In contrast, no increase in Na-APR-1 specific T cell responses were induced by the vaccine. Mass cytometry revealed that, Na-GST-1 cytokine producing CD4$^+$ T cells were CD161$^+$ memory cells expressing CTLA-4 and CD40-L. Blocking CTLA-4 enhanced the cytokine response to Na-GST-1.

**Funding:** This project has received funding from: 1- The European Union's Seventh Framework Programme for research, technological development and demonstration (https://cordis.europa.eu/programme/id/FP7) under grant agreement no 602843 (HOOKVAC) to Remko Van Leeuwen. 2- The German Center for Infection Research DZIF CRG (https://www.dzif.de/en) 80207STRUK (TTU.03.703) to AAA. 3- Bontius Stichting (https://www.cbf.nl/organisatie/bontius-stichting–research-foundation), to MY. 4- Stichting Tabernaleporis (https://www.tabernaleporis.nl/), to MY. 5- Leiden University Medical Center Strategic Fund for Leiden Controlled Human Infection Center (L-CHIC), to MY. The funders had no role in study design, data collection and analysis, decision to publish or preparation of the manuscript.

**Competing interests:** no authors have competing interests.

In Gabonese volunteers, hookworm vaccine candidate, Na-GST-1, induces detectable CD4$^+$ T cell responses that correlate with specific antibody levels. As these CD4$^+$ T cells express CTLA-4, and blocking this inhibitory molecules resulted in enhanced cytokine production, the question arises whether this pathway can be targeted to enhance vaccine immunogenicity.

## Author summary

Two hookworm vaccine candidate (Na-GST-1 and Na-APR-1) have been tested in Gabonese and found to be safe and to induce antibody response. We aimed to study the cellular immune responses among vaccinated and unvaccinated volunteers. We found that Na-GST-1 induced CD4$^+$ T cell responses (IL-2, TNF) among the vaccinated volunteers that received the high vaccine dose (100 ug). Furthermore Na-GST-1 specific memory T cells were found to express the inhibitory molecule CTLA-4. These responses was not observed in those who received the low dose of the Na-GST-1 vaccine, or those who received Na-APR-1 or HBV. By blocking CTLA-4, we observed an increase in TNF production. Our data suggest that an intervention involving blockage of the CTLA-4 molecule in the vaccinated could be beneficial in endemic settings where vaccine responses have been shown to be lower compared to non-endemic settings.

## Introduction

Human hookworm infection affects approximately 740 million people worldwide and, 85% of cases are caused by *Necator americanus* (*N. americanus*), while the remaining are accounted by *Ancylostoma duodenale* (*A. duodenale*) [1, 2]. Most people with light infections experience mild diarrhea and abdominal pain, while more severe and chronic infections may cause blood loss leading to severe anemia and hypoproteinaemia [3], compromising the health of children and pregnant women [3]. Currently, the main approach to control hookworm infections worldwide, is the annual Mass Drug Administration (MDA) with anti-helminthic drugs, mebendazole or albendazole. However, this approach has not led to a successful control or elimination of hookworm infections [4, 5] due to rapid reinfection in areas where transmission cannot be interrupted [6]. Therefore, to achieve effective control and elimination, it is important to develop an effective vaccine for human hookworm disease to complement MDA and other public health measures.

The proof of concept that hookworm vaccines can be effective comes from the field of veterinary medicine where a radiation-attenuated *A. caninum* infective larval stage (L3) vaccine was shown to offer high protection against canine hookworm infection [7]. Subsequent efforts have been directed towards discovery of L3-stage antigens that could mediate protective immunity in humans [8, 9] with the identification of L3-stage Na-ASP-2 protein as a potential vaccine candidate. Na-ASP-2 was shown to be safe and immunogenic in volunteers from non-endemic areas without history of hookworm infection [9]. However, when tested in an endemic area in Brazil it caused generalized IgE- mediated urticaria leading to the termination of the Phase 1 clinical trial [10]. Subsequent approaches targeted hidden proteins, less likely to induce sensitization during natural hookworm infection, such as enzymes involved in the blood-feeding process of adult worms [11, 12]. This led to the selection of the glutathione S-transferase (GST) and the aspartic protease (APR) hemoglobinases [13] as vaccine candidates. It was postulated that Na-GST-1 and Na-APR-1-specific antibodies would neutralize the function of these enzymes, leading to

interruption of blood digestion and ultimately causing, parasite death. The Na-GST-1 vaccine candidate has already been tested in Phase 1 trials in hookworm-naive (in USA) and hookworm-exposed (in Brazil) adult volunteers [14] and was found to be safe and immunogenic leading to the induction of antigen-specific IgG antibodies in a dose-dependent manner [14].

To date, only limited data are available on the human cellular immune responses during hookworm infection [15–17] and none when testing hookworm vaccines candidates.

In order to understand the immunogenicity of a vaccine and to identify correlates of protection, it is important to characterize ex vivo vaccine- induced cellular as well as humoral responses [18]. The efficacy of most preventive vaccines relies on antibody responses [18] and in this context, activation of T-helper cells is required for the induction of high-affinity antibodies [18–20]. Therefore, we characterized intracellular cytokine production before and after vaccination of Gabonese volunteers with Na-GST-1 and Na-APR-1, by flow and mass cytometry following in vitro stimulation with vaccine candidates.

## Materials and methods

### Ethics statement

The trial was approved by the National Ethics Committee of Gabon (reference number 0033/2014/SG/CNE) and was conducted under an Investigational New Drug (IND) application to the US Food and Drug Administration (ClinicalTrials.gov Identifier: NCT02126462). The trial followed Good Clinical Practice and Good Clinical and Laboratory Practice as defined by the International Conference on Harmonization (ICH E6[R2])

### Study population

This study was conducted at the "Centre de Recherches Médicales de Lambaréné" (CERMEL) in Lambaréné, a semi-urban municipality in Gabon (central Africa), where hookworm infection is endemic. Healthy adults aged 18–50 years living in Lambaréné and the surrounding areas were recruited. Prior to inclusion a questionnaire to evaluate study comprehension was administered to the participants and written informed consent was obtained.

At baseline, the general health status of each participant was assessed by physical examination and laboratory tests. Participants with systemic disease, HIV or chronic hepatitis B/C infection, under corticosteroids or immunosuppressive treatment, having received a live vaccine within the previous month, or currently pregnant or breastfeeding were excluded. If at screening, helminth infections were detected by fecal (*Ascaris lumbricoides*, *Trichuris trichiura*, *Necator americanus*, *Ancylostoma duodenale* and *Strongyloides stercoralis*) and urine (*Schistosoma hematobium*) microscopy they were treated with either albendazole or praziquantel, respectively, at least two weeks before inclusion into the trial. Female participants included were asked to use reliable contraception up to one month after the last vaccination.

### Phase 1 trial of Na-GST-1 and Na-APR-1

This study was a randomized double-blind controlled Phase 1 clinical trial of the Na-GST-1 and the Na-APR-1 hookworm vaccines. In total, 32 participants were enrolled in two cohorts of 16 participants each. The 16 participants included 12 participants vaccinated with both recombinant, Na-GST-1 and Na-APR-1 (M74), plus 5μg GLA-AF (experimental group) and 4 participants vaccinated with hepatitis B vaccine (HBV) plus a buffer saline solution (control group). It has to be noted that Na-APR-1 (M74) is a catalytically inactive mutant protein that was designed to improve the stability of Na-APR-1 by genetically mutating two aspartic acid residues to alanine. Na-APR-1 (M74) induced antibodies that reacted to wild type Na-APR-1.

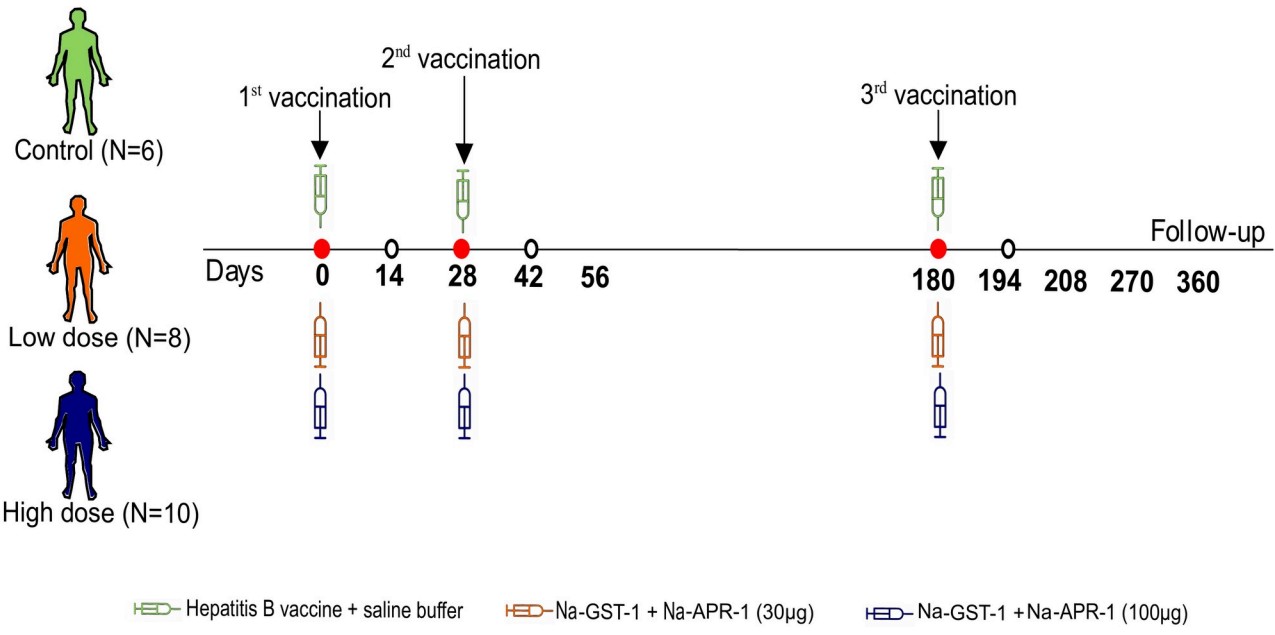

**Fig 1. Vaccination and sampling time points.**

Vaccinations were delivered by intramuscular injection in the deltoid muscle. Each cohort was distinguished by the vaccine dose, namely low dose (30 µg) for group 1 and high dose (100 µg) for group 2. Participants received one dose of each vaccine tested, one in each upper arm. In the control group participants received HBV in one upper arm and saline in the other. Participants were vaccinated at days 0, 28 and 180 as detailed by Adegnika et al. [21] and in S1 Fig. Heparinized blood samples were collected before each vaccination at days 0, 28, 180, and at days 14, 42, 56, 194, 208, 270 and 360 following vaccination. For the immunogenicity assay days 0, 14, 28, 42, 180 and 194 were selected (Fig 1).

## Antibody measurement

The IgG antibody levels to Na-GST-1 was measured as described previously in great detail [21]. The mean of the optical density at 492 nm of each test sera was interpolated onto the standard calibration curve of reference serum to derive the arbitrary units (AU) of anti-Na-GST-1 or anti-Na-APR-1(M74) IgG [10, 22].

## Cell isolation and storage

Peripheral blood mononuclear cells (PBMC) were isolated from heparinized venous blood within 8 hours after venipuncture by density gradient centrifugation over Ficoll-Hypaque (Apotheek AZL, Leiden, The Netherlands) as described previously [23]. Isolated PBMCs were cryopreserved in RPMI 20% FCS 10% DMSO (Merk, cat: 67685) using a controlled-grade Mr. Frosty Freezing Container (Thermo Scientific, cat: 5100–0001) containing isopropyl alcohol, and stored overnight at -80˚C prior to transfer into liquid nitrogen for long-term storage.

## PBMC in vitro stimulation

PBMCs were thawed and rested overnight prior to in vitro stimulation with vaccine candidate antigens. All samples met the minimum criteria of >80% viability (trypan blue staining) after thawing and resting.

PBMCs were cultured in 200µl RPMI 1640 (Gibco, Invitrogen, Carlsbad, CA, USA) supplemented with 10% FCS (Greiner Bio-One GmbH, Frickenhausen, Germany), 100 U/ml penicillin (Astellas, Tokyo, Japan), 10 µg/ml streptomycin, 1mM pyruvate and 2mM L-glutamine (all from Sigma-Aldrich, CA, USA). Five hundred thousand cells per well were stimulated in sterile polypropylene U-bottom microtiter plates with medium (negative control), 1ug/ml Na-GST-1, 1ug/ml Na-APR-1 (M74) or 200 ng/ml staphylococcus enterotoxin B (SEB) (positive control), all in the presence of co-stimulatory antibodies CD28/CD49d (BD Biosciences cat: 555725/555501) both at a concentration of 2µg/ml. Brefeldin A (10µg/ml, Sigma; cat: B7651) was added to the cells culture 5 hours post- stimulation and cells were cultured for a total of 24 hours at 37˚C in the presence of 5% $CO_2$.

## Immunophenotyping assays

**Flow cytometry (FACS) intracellular cytokine staining.** After in vitro stimulation, cells were directly stained with LIVE/DEAD Fixable Aqua dead cell Stain Kit (Invitrogen, cat: L34957) for 15 min in the dark at room temperature (RT), followed by 15 min fixation with 1.9% PFA in the dark at RT. Thereafter cells were stained for 30 min at 4˚C with a cocktail of CD3-APC-eF780, CD4-PerCP-eF710, IFN-γ-BV-421, IL-2-FITC, IL-4/-5/-13-PE, IL-10-APC and TNF-α-PE-Cy7 antibodies (for dilutions and antibody details see S1 Table) prepared in permeabilization buffer (eBioscience, cat: 88-8824-00). Cells were acquired on BD FACSCanto II flow cytometer (BD Biosciences) and analyzed using FlowJo V10 (TreeStar, San Carlos, CA). CD4+ T cell producing cytokines were selected using the gating strategy shown in S2A Fig. Two subjects, one in the low dose group (30 µg) and one in the high dose (100 µg), were excluded from immunogenicity data analysis due to high spontaneous production of cytokines in culture medium.

**Mass cytometry (CyTOF) intracellular analysis.** Na-GST-1 stimulated PBMC (~3 millions of cells) were incubated with 1 mL of 500 µM Cell-ID intercalator-103Rh (Fluidigm, cat: 201103A; 500x dilution), for 15 min at RT to identify dead cells. Next, cells were incubated for 10 min at RT with 50 µL of human TruStain FcX Fc-receptor blocking solution (Biolegend, San Diego, CA, USA; 10x dilution), and were subsequently stained with 50µL of a surface antibody cocktail (cocktail 1), freshly prepared (S2 Table) for 45 min at RT prior to fixation with 1mL of 1x MaxPar Fix I buffer (Fluidigm, cat: 201065) for 20 minutes at RT. Additionally, cells were stained a second time for 30 minutes at RT with 50 µl of an intracellular antibody cocktail (cocktail 2) freshly prepared (S2 Table) in permeabilization buffer (Fluidigm, cat: 201066) and followed by overnight staining at 4˚C with 1 mL of 125 µM Cell-ID Intercalator-Ir (Fluidigm, cat: 201192A; 1000x dilution) in MaxPar Fix and Perm buffer (Fluidigm, cat: 201067). Cells acquisition was performed using Helios mass cytometer (Fluidigm, South San Francisco, CA, USA) at a concentration of $1x10^6$ cells/mL in Milli-Q water, filtered using cells-strainer and complemented with 10% EQ Four Element Calibration Beads (Fluidigm, cat: 201078). In addition to the antibody panel detection channels (including intercalator ones), calibration beads (140Ce, 151Eu, 153Eu, 165Ho and 175Lu) and contamination (133Cd, 138Ba, 208Pb) channels were activated. After cell acquisition, data from the calibration beads were used to normalize signal fluctuations. The normalized FCS files were exported and analyzed using FlowJo V10 (TreeStar, San Carlos, CA) in order to gate out EQ beads and select live CD45+ cells (S4 Fig). The FCS files from selected live CD45+ cells were then analyzed using the hierarchical stochastic neighbor embedding (HSNE) [24–26], a dimensionality reduction visualization tool, to identify CD4+ T cells responding to Na-GST-1 stimulation (S4 and S5 Figs).

## CTLA-4 blocking assay

PBMC were incubated in culture medium (RPMI 10%FCS), with 3 µg/ml CTLA-4 blocking antibody (BPS Bioscience, cat: 71212) or IgG1 isotype (Invitrogen, cat: 16-4714-85), at a

concentration of $5 \times 10^5$ cells/well, for 1hr at 37˚C, 5% $CO_2$ in sterile polypropylene U-bottom microtiter plates control. Next, cells were stimulated for 24 hours with either 1 µg/mL Na-GST-1, or culture medium only, as negative control, in the presence co-stimulatory antibodies CD28/CD49d (BD Biosciences cat: 555725/555501), both at a final concentration of 2µg/ml. Supernatants were collected to measure tumor necrosis factor (TNF) by enzyme-linked immunosorbent assay (ELISA) (BD Biosciences, cat: 555212), following manufacturer's recommendation. It has to be noted that it was not possible to show an increase in cytokine production when intracellular cytokine staining was used.

## Statistical analysis

For each stimulation condition, the percentages of CD4+ T cells producing each cytokine or cytokine combination were determined. The response to stimulation with medium was subtracted from the Ag stimulated responses to give us antigen specific percentage of cytokine producing cells. Linear mixed-effects models [27] were used to assess the change in cytokine producing CD4+ T cells over time, in the vaccinated groups and control. The participant ID was modeled as random effect, the time as fixed effect and the percentage for cytokine producing cells as the dependent variable. To assess the change between the frequency of CD4+ T cells producing one or more cytokines at day 194 compared to baseline, pairwise comparisons were performed using Wilcoxon paired test. The statistical test was one-tailed with significance level set at $p \leq 0.05$. Spearman's correlation was used to determine the correlation between percentages of CD4+ T cells producing cytokines and the antibody response. Kruskal Wallis test was used to compare TNF production following anti-CTLA-4 blocking with isotype and medium conditions. All statistical analysis were performed in R software [28].

## Results

### Characteristics of study participants

Out of 32 participants enrolled in the trial, 24 completed the full vaccination schedule and were included in the analysis of the responses to hookworm vaccine candidates (S1 Fig). These were 6 participants in the control group (HBV vaccine), 8 participants in the low dose group (30 µg Na-GST + 30 µg Na-APR) and 10 participants in the high dose group (100 µg Na-GST + 100 µg Na-APR). The characteristics of the study participants are given in Table 1. There were no significant difference between the groups in terms of age, sex, body mass index (BMI), living area, and helminth infection status.

### CD4+ T cell responses to Na-GST-1 and Na-APR -1 following immunization

At baseline, there were no detectable cytokine producing CD4+ T cells in response to Na-GST-1, there were to Na-APR-1 (S3 Fig). Following vaccination, in the high dose group (100 µg), CD4+ T cells expressing IL-2 (p = 0.048) or TNF (p = 0.013) (Fig 2A), as well as co-expressing TNF and IL-2 (p = 0.0156) (Fig 2B), increased significantly, in response to Na-GST-1, from day 0 (before vaccination) up to day 194 (two weeks after the third vaccination). No significant changes over time were detected for IFN-γ, $T_H2$ cytokines (IL-4/-5/-13) or IL-10 in response to Na-GST-1. No significant increase over time, compared to baseline (day 0), were detected when considering low dose Na-GST-1 vaccinated group (Fig 2A). With respect to Na-APR-1, there was no significant increase in cytokine producing CD4+ T cells from baseline (day 0) up to day 194.

**Table 1. Characteristics of study participants (N = 24).**

| | HBV (control) | Low dose (30µg) | High dose (100µg) | P-value |
|---|---|---|---|---|
| N | 6 | 8 | 10 | |
| Age years; mean (range) | 21.5 (19–24) | 29 (18–50) | 21.1 (19–44) | 0.077[b] |
| BMI (mean± SD) | 21.8 ± 2.4 | 21.5 ± 8.8 | 21.8 ±7.9 | 0.297[b] |
| Sex female; | | | | |
| Female; n(%) | 1 (16.7%) | 3 (37.5%) | 3 (30%) | 0.706[a] |
| Living Area | | | | |
| Rural; n(%) | 4 (72.7%) | 7 (87.5%) | 5 (50%) | 0.260[a] |
| Helminth infection | | | | |
| Any helminth n (%) | 2 (33.3%) | 7 (77.8%) | 7 (70%) | 0.110[a] |
| *Ascaris lumbricoides* | 0/6 | 1/8 | 2/10 | |
| *Trichuris trichiura* | 1/6 | 0/8 | 2/10 | |
| Hookworm | 0/6 | 2/8 | 1/10 | |
| *Strongoloides stercoralis* | 0/6 | 0/8 | 2/10 | |
| *Schistosoma haematobium* | 1/6 | 5/8 | 4/10 | |
| Other | 0/6 | 0/8 | 1/10* | |

[a]Chi-Square

[b]KruskalWallis

* *Tenia sp*

The absence of cytokine production in response to Na-GST-1 in the HBV vaccinated control group, as well as the lack of any differences seen in response to SEB stimulation over time (S2B Fig) indicates that Na-GST-1 vaccination leads to antigen specific T cell responses.

## Correlation between IgG antibody titers and cytokine responses to Na-GST-1

Na-GST-1 specific IgG antibody level [21] was shown to be highest in the high dose group after the third vaccination on day 194. A significant positive correlation (p = 0.003, rho = 0.83) was seen between CD4$^+$ T cells secreting TNF and Na-GST-1 specific IgG on day 194 (Fig 2C). No correlation was found between IL-2 producing, or IL-2 and TNF co-producing CD4$^+$ T cells and antigen specific IgG (S2C Fig).

## CTLA-4 expression on Na-GST-1 -specific CD4$^+$ T cells

We next, examined in detail, the phenotype of these Na-GST-1 -specific CD4$^+$ T cells in 3 donors, by mass cytometry. To this end, a 37-metal isotope-tagged monoclonal antibody panel for mass cytometry was used (S2 Table). Hierarchical stochastic neighbor embedding (HSNE) [24] plots in Fig 3A, are depicting the cellular heterogeneity of Na-GST-1 specific CD4$^+$ T cell population, and identifying five subsets. Within the five distinct subsets (Fig 3A), the frequency of TNF and IL-2 producing cells in Na-GST-1 stimulated cells compared to medium, was high in subset 5, for all donors (S5B Fig). Subset 5 comprises of CD161$^+$ CD4$^+$ memory T cells (Fig 3A). Further analysis of this subset revealed 15 phenotypically distinct clusters, with the markers depicted in the heatmap (Fig 3B). The TNF and IL-2 producing cells in response to antigen were largely in cluster 15 in all donors (Figs 3C and S5C). Cluster 15 is defined by the expression of CD40-L and CTLA-4, within the CD161$^+$ CD4$^+$ memory T cell compartment (Fig 3D). In this cluster, TNF was found to be the main cytokine produced in response to Na-GST-1 stimulation, and to a lesser extent IL-2 (S5C Fig).

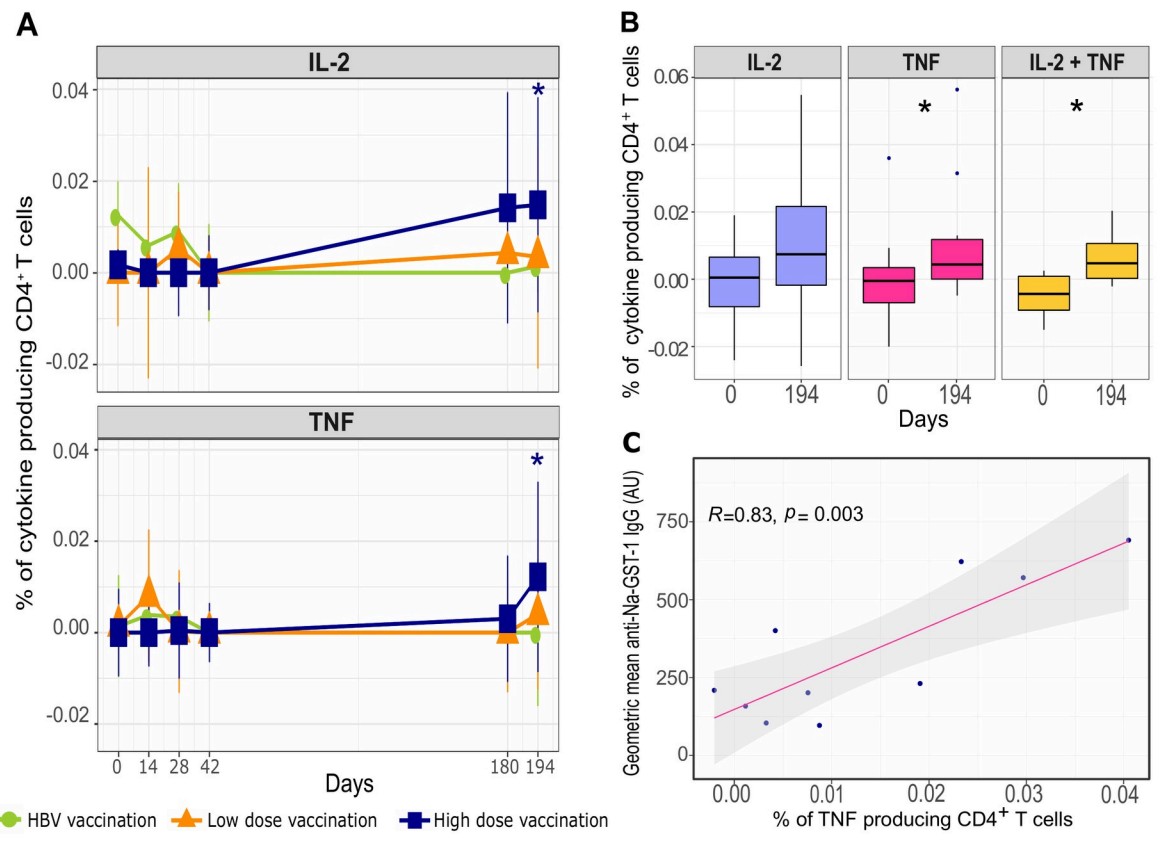

**Fig 2. Cytokine responses to Na-GST-1 vaccination. (A)** The frequency of IL-2 and TNF producing CD4+ T cells in response to Na-GST-1 stimulation, over time compared to baseline (day 0), in control (green line), low dose (orange line) and high dose group (blue line), using linear mixed model for statistical analysis. Cells producing cytokines are expressed as percentage of CD4+ T cells. The mean and standard deviation of cytokine producing cells is given for each time point and each vaccine group. **(B)** Frequency of CD4+ T cells producing IL-2 or TNF alone or together in response to Na-GST-1 on day 0 and day 194, in the high dose (100 μg) group. Data are presented as boxplots representing the median, 1st and 3rd quantile. Whiskers are extending to the maximum/minimum, no further than 1.5x the IQR (interquartile range). Wilcoxon one-tailed paired test was performed for comparison between day 0 and day 194. **(C)** Correlation between the change induced by vaccination in Na-GST-1 specific IgG antibody levels and the frequency of TNF producing CD4+ T cells, in the high dose group (Spearman correlation test rho = 0.83, p = 0.003). The change in IgG and cytokine producing cells was determine by subtracting the baseline value (day 0) from day 194 value. The antibody levels are given as arbitrary unit (AU). The frequency of cytokine producing cells in (A), (B) and (C) was determined as percentage of total number of CD4+ T cell. (*) indicates the significance p≤0.05 in (A) and (B).

## CTLA-4 blockade in vitro enhances Na-GST-1 antigen-specific TNF production

As CTLA-4 is known to be an inhibitory molecule [29], the question whether CTLA-4 can result in dampening the cytokine response of antigen stimulated CD4+ T cells was examined next. PBMC of 6 donors from the high dose vaccinated group, taken at pre and at day 194 post vaccination, were stimulated with Na-GST-1 in the presence and the absence of anti-CTLA-4 blocking antibodies. The increase from day 0 to day 194 in TNF production after anti- CTLA-4 antibody was used, was significantly more than the increase seen with control antibodies or medium (p = 0.003) (Fig 4).

## Discussion

This is the first study to examine T cell responses following vaccination with hookworm vaccine candidates Na-GST-1 and Na-APR-1, which was assessed in a population residing in a

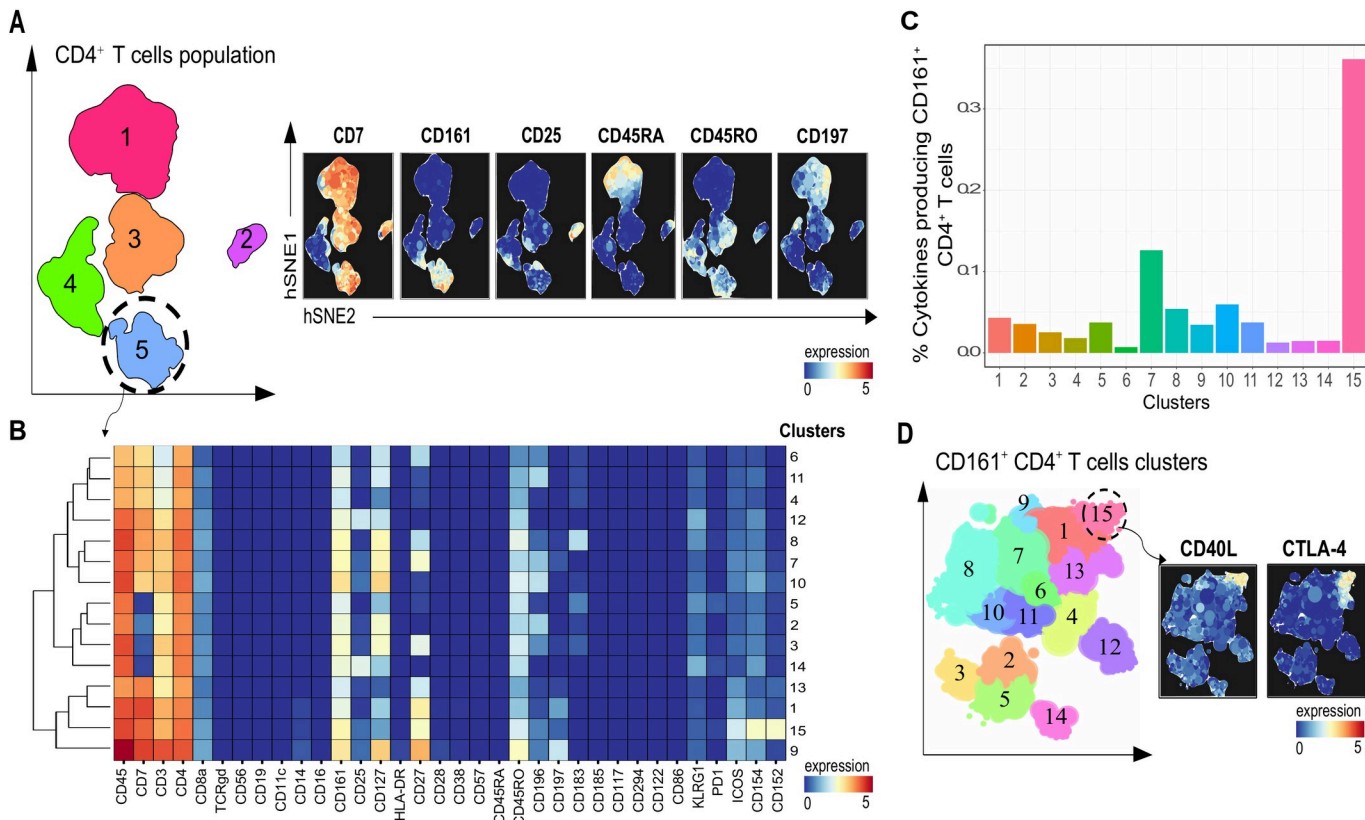

**Fig 3. Na-GST-1 specific cell phenotype. (A)** HSNE embedding of 1.1 million CD4+ T cells at day 194 in Na-GST-1 high dose (100 μg) recipients (n = 3), depicting 5 subsets. The colour represents arcsin5- transformed expression values of indicated markers. **(B)** Heatmap summarizing the median expression of markers present on identified clusters of CD4+ CD161+ T cell subset (subset 5). The colours are the same as for the HSNE plots in A. **(C)** Bar plots depicting the frequency of clusters producing cytokines relative to CD4+ CD161+ T cells. **(D)** Cluster partitioning of CD4+ CD161+ T cells and HSNE plots highlighting cluster 15, which has an increased expression of both CD40-L and CTLA-4.

hookworm endemic area in Africa. Both vaccine candidates have been reported to be safe and immunogenic [21]. Here, we characterized T cells immune response to both, Na-GST-1 and Na-APR-1 vaccine candidates. There was no response at baseline to Na-GST-1, but following vaccination, T cell responses became detectable. This was different from Na-APR-1, to which cytokine (TNF) producing T cells were detectable before vaccination but did not increase fol-lowing vaccination. The fact that already detectable Na-APR-1 response was not boosted by vaccination, might be due to the low sensitivity of intracellular cytokine staining [30] that does not allow the detection of an incremental increase in the response. It might also reflect the effect of pre-existing antibodies on vaccine responses, often described to interfere with live vaccine efficacy [31]. In any case, it is in line with the lower vaccine induced antibody response to Na-APR-1 compared to Na-GST-1 [21]. The question remains This might whether either higher doses of Na-APR-1 or a different adjuvant would help to get a better response.

Na-GST-1 -specific IL-2 and TNF producing CD4+ T cells were increased in frequency after the third vaccination (day 194), in the group receiving the high dose vaccine. We did not find antigen- specific IFN-γ production following vaccination. This seems to also be the case during natural hookworm infection where TNF production in response to antigen was shown to be elevated in infected subjects, in contrast to IFN-γ [32, 33], and was negatively correlated with egg burden [33], suggesting a role for TNF production in natural immunity to hookworm infection.

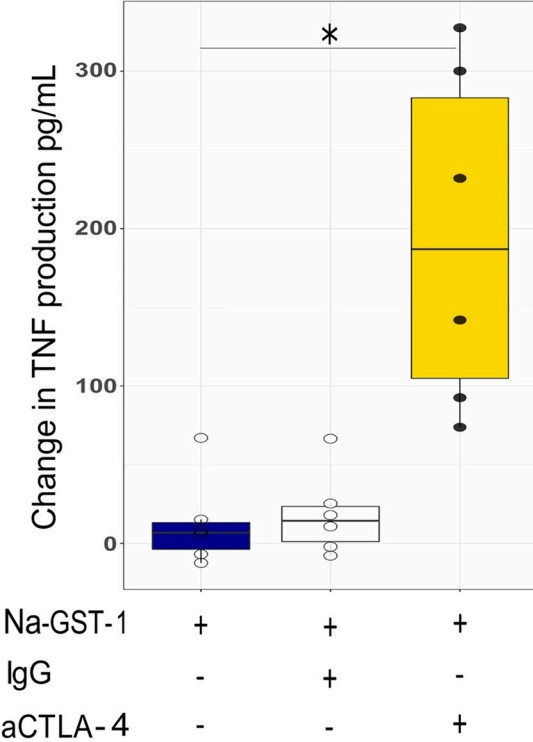

**Fig 4. Antigen specific cytokine production following blocking of CTLA-4.** Analysis of TNF production in response to Na-GST-1 in cells treated with anti-CTLA-4. Data are presented as boxplots showing the median and IQR. Whiskers are extending to the maximum/minimum, no further than 1.5x the IQR. Individual values are shown as points. The change in TNF production in response to Na-GST-1 was determine by subtracting baseline (day 0) value from day 194 value when either medium, control IgG or anti CTLA-4 was used. A significant increase in TNF production was observed after CTLA-4 blocking compared to IgG isotype or medium control. Statistical significance was determined using Kruskal Wallis test. (*) indicates the significance p≤0.05.

In some other vaccination studies, the prominence of TNF rather than IFN-γ response has been reported [34, 35]. Data from a study in healthy Brazilian volunteers vaccinated with a *Schistosoma mansoni* vaccine candidate Sm14, also showed an increase in CD4[+] T cells producing TNF and IL-2, 30 days after the third vaccination, with little IFN-γ detected [34]. Similarly, studies assessing cytokine responses in subjects immunized with vaccines such as hepatitis B, tetanus toxoid or malaria (RTS'S/AS and AMA-1), have shown that TNF and IL-2 are the main cytokines detected following immunization [35–38]. Moreover, we show an increase in TNF and IL-2 co-expressing CD4[+] T cells upon vaccination, which might be important, as vaccine-induced CD4[+] T cells producing more than one cytokine have been shown to correlate with protection in several vaccine studies [39–41].

We found a positive correlation between antibody levels and Na-GST-1 specific TNF producing CD4[+] T cells, two weeks after the third vaccination (day 194). This could suggest that TNF secretion is involved in antibody production as previous studies have shown antibody responses to vaccines are compromised in patients treated with TNF blockade [42–44].

Characterization in more detail of the phenotype of antigen stimulated CD4[+] T cells producing cytokines in response to Na-GST-1 showed that the cells had a memory phenotype and expressed CD40-L which indicate stimulation through TCR signaling [45, 46]. Moreover, these antigen responding cells, expressed CD161, a marker that characterizes CD4[+] T cells with potent functional capacity [47]. In addition, CTLA-4, which negatively regulates T-cell

responses [48], was expressed by *Na*-GST-1 responsive cells. Blocking of CTLA-4 with a neutralizing antibody resulted in enhanced TNF production upon in vitro stimulation. The question remains as to whether the expression of CTLA-4 on Na-GST-1 specific CD4+ T cells is characteristic of cells from vaccinated subjects that reside in areas endemic for hookworm infection or whether the same is true for vaccinated subjects from North America with no prior exposure to hookworms [14]. This might be relevant to the observations that some vaccines show poor immunogenicity or efficacy in Africa [49, 50]. However, comparing antibody responses between North Americans [14] and Gabonese [21] vaccinated with Na-GST-1 and Na-APR-1, does not seem to show large differences in antibody levels. Therefore, further studies on quality of the antibodies are needed before we consider the relevance of CTLA-4 expression on CD4+ T cells induced upon vaccination in Gabonese volunteers.

In summary, we were able to characterize CD4+ T cell responses to vaccination with Na-GST-1, showing a prominent TNF production by antigen stimulated cells that correlated with antibody production. The cells were characterized by expression of CD45RO, CD161, CD40L and CTLA-4. Blocking of CTLA-4 enhanced the production of TNF. Whether CTLA-4 expression restricts immunogenicity or functionality of the vaccine candidates will need to be investigated further.

## Supporting information

**S1 Table. Flow cytometry antibody panel.**
(DOCX)

**S2 Table. Mass cytometry (CyTOF) antibody panel.**
(DOCX)

**S1 Fig. Study flow chart.**
(TIF)

**S2 Fig.** (**A**) Flow cytometry gating strategy of CD4+ T cells for one representative sample after SEB stimulation. Cells were first gated on time or Forward scatter (FSC)-W and FSC-H to select singlets. Thereafter, cells were gated on viability dye and FSC-A to select live cells. Subsequently, cells producing IL-2, TNF, IFN-γ, IL-10 and the $T_H2$ cytokines (IL-4, IL-5 and IL-13) were gated from cells expressing CD3 and CD4. (**B**) Cytokine response to SEB over time. Boxplots representing the median, 1st and 3rd quantile. Whiskers are extending to the maximum/minimum, no further than 1.5x the IQR. All points are shown. (**C**) Spearman correlation between the change (day 194 –day 0) induced by vaccination in Na-GST-1 specific IgG antibody levels, given in arbitrary units (AU), and the frequency of IL-2 producing CD4+ T cells, in the high dose group. Solid line indicates predicted values from linear regression analysis with the shaded band showing the 95% confidence interval.
(TIF)

**S3 Fig.** The frequency of IL-2 and TNF producing CD4+ T cells in response to Na-GST-1 or Na-APR-1 stimulation compared to medium at baseline (day 0), in control (green line), low dose (orange line) and high dose group (blue line). Cells producing cytokines are expressed as percentage of CD4+ T cells. Each line represents an individual donor. Wilcoxon one-tailed paired test was performed for comparison between medium and Na-GST-1 or Na-APR-1 stimulation. (*) indicates the significance p≤0.05 in (A) and (B).
(TIF)

**S4 Fig. Single live CD45+ cells were manually gated in stimulated and unstimulated samples using the Gaussian parameter (Residual, Center, Offset and Width).** Unsupervised

analysis was performed using HSNE on Cytosplore to visualize the relative distribution of cell populations within the CD45$^+$ cell. At the first level HSNE embedding was used where CD45 + cells clustered based on surface markers expression, identifying the major immune lineages were identified.
(TIF)

**S5 Fig.** (**A**) HSNE embedding depicting the marker expression of the major immune lineages identified within CD45$^+$ cells. The colour represents the arcsin5- transformed expression values of indicated markers. (**B**) Cytokines producing CD4$^+$ T cell populations in Na-GST-1 stimulated cells. The frequency of cytokine producing cells is of total number of CD4$^+$ T cells. (**C**) Cytokines producing CD161$^+$ CD4$^+$ T cell population in Na-GST-1 stimulated cells in all donors. The frequency of cytokine producing is of total number of CD161 expressing CD4$^+$ T cell. Both medium (MED) and Na-GST-1 (GST) stimulation is shown and lines depict the same donors in (**B**) and (**C**).
(TIF)

## Acknowledgments

We thank all the participants who volunteered to be included in this study and the entire clinical team of CERMEL, particularly Loretta Issanga Mabicka, Rodrigue Bikangui, and Jo-Lewis Banga Ndzouboukou for the collection of the study samples. We also thank Stephen C. De Rosa and Sanne de Jong for technical advice.

## Author Contributions

**Conceptualization:** Martin P. Grobusch, Marguerite Massinga Loembe, Maria Yazdanbakhsh.

**Data curation:** Yoanne D. Mouwenda, Koen A. Stam.

**Formal analysis:** Yoanne D. Mouwenda, Koen A. Stam.

**Funding acquisition:** Ayola A. Adegnika, Maria Yazdanbakhsh.

**Investigation:** Yoanne D. Mouwenda, Madeleine E. Betouke Ongwe, Friederike Sonnet, Lucja A. Labuda, Sophie De Vries, Frejus J. Zinsou, Yabo J. Honkpehedji, Jean-Claude Dejon Agobe, Simon P. Jochems, Ayola A. Adegnika.

**Methodology:** Martin P. Grobusch, David J. Diemert, Remko van Leeuwen, Maria E. Bottazzi, Peter J. Hotez, Jeffrey M. Bethony, Ayola A. Adegnika.

**Project administration:** Yoanne D. Mouwenda, Maria Yazdanbakhsh.

**Resources:** Martin P. Grobusch, David J. Diemert, Ayola A. Adegnika.

**Software:** Yoanne D. Mouwenda, Koen A. Stam.

**Supervision:** Peter G. Kremsner, Simon P. Jochems, Marguerite Massinga Loembe, Maria Yazdanbakhsh.

**Validation:** Yoanne D. Mouwenda, Maria Yazdanbakhsh.

**Visualization:** Yoanne D. Mouwenda, Koen A. Stam.

**Writing – original draft:** Yoanne D. Mouwenda, Maria Yazdanbakhsh.

**Writing – review & editing:** Yoanne D. Mouwenda, Madeleine E. Betouke Ongwe, Friederike Sonnet, Koen A. Stam, Lucja A. Labuda, Sophie De Vries, Martin P. Grobusch, Frejus J.

Zinsou, Yabo J. Honkpehedji, Jean-Claude Dejon Agobe, David J. Diemert, Remko van Leeuwen, Maria E. Bottazzi, Peter J. Hotez, Peter G. Kremsner, Jeffrey M. Bethony, Simon P. Jochems, Ayola A. Adegnika, Marguerite Massinga Loembe, Maria Yazdanbakhsh.

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
