## [Decision Letter · Decision Letter 0]

1 Jul 2021

Dear Mrs Mouwenda,

Thank you very much for submitting your manuscript "Characterization of T cell responses to co-administered hookworm vaccine candidates Na-GST-1 and Na-APR-1 in healthy adults in Gabon" for consideration at PLOS Neglected Tropical Diseases. As with all papers reviewed by the journal, your manuscript was reviewed by members of the editorial board and by several independent reviewers. The reviewers appreciated the attention to an important topic. Based on the reviews, we are likely to accept this manuscript for publication, providing that you modify the manuscript according to the review recommendations. 

Sincerely,

Keke C Fairfax, PhD

Deputy Editor

Keke Fairfax

Deputy Editor

Reviewer's Responses to Questions

**Key Review Criteria Required for Acceptance?**

**Methods**

-Are the objectives of the study clearly articulated with a clear testable hypothesis stated?

-Is the study design appropriate to address the stated objectives?

-Is the population clearly described and appropriate for the hypothesis being tested?

-Is the sample size sufficient to ensure adequate power to address the hypothesis being tested?

-Were correct statistical analysis used to support conclusions?

-Are there concerns about ethical or regulatory requirements being met?

Reviewer #1: Methods: Sufficient information should be provided on subject selection an immunization without having to read reference 21. A flow diagram would be useful showing randomizations (N=32) and losses to follow-up yielding number finally analyzed (n=24). Figure 1, although useful, is not helpful in this respect because it does not show randomizations (allocated to what?) and shows the number followed-up rather than the number randomized with the final ‘3-group’ composition (presumably the groups to which subjects were randomized?). 

Methods: I am curious why the subjects were vaccinated intramuscularly in the forearm (line 129) rather than the upper arm where the deltoid muscle is found?

Methods (line 141): What is ‘(M74)’?

Methods (line 148): did the Frosty™ Freezing Container contain isopropyl alcohol?

Methods (line 207): 5*105 = 5x105?

Methods (line 209): 500x diluted?

Methods (statistical analysis) – this needs to be edited carefully for clarity

Reviewer #2: The methods are sound and well written. There is no ethics or regulatory concerns.

**Results**

-Does the analysis presented match the analysis plan?

-Are the results clearly and completely presented?

-Are the figures (Tables, Images) of sufficient quality for clarity?

Reviewer #1: (No Response)

Reviewer #2: The paper is overall clear and well written, but sometimes lack in details in the description of results. 

For example regarding Figure 2, the authors state : "Following vaccination Na-GST-1-specific responses were detectable, but Na-APR-1-specific responses were not.” This is a broad statement, especially as the first sentence of the paragraph. Could the author better explain their results? Is it that restimulation with Na-APR-1 did not give any T cell response at all, or no specific cytokine response. which cytokines have been tested? Here the negative result is actually quite important and should be further defined. Have the authors considered measuring total cytokines released in the medium after stimulation, as it might be more sensitive that intracellular detection.

Figure 2 legend : Figure 2, in the legend on the figure orange is indicated as GST 30 ug and blue for GST 100ug. To my understanding all participants have been immunized with both vaccine candidates (Na-GST-1 and Na-APR-1). It is unclear thus if the color refer to the immunisations or whether it refer to restimulation of the PBMC in vitro.

Lines246-247 , the authors state “No significant changes were detected when considering low dose Na-GST-1vaccinated group (Fig 2A)”. Is it compared to day 0 (baseline) or to the other groups (HBV or NA-GST-1 100ug).

Line 289 : There is no letter D in Figure S4

Fig 3 D while quite standard, really poor choice of color for the level of expression as the cluster 15 appears in white and can barely be seen in the image. Maybe the authors could use black rather than white in the middle of the gradient to facilitate visualization of the results.

Figure S4 The mass cytometry panel is quite comprehensive and allow detection of numerous populations. While the authors might want to keep some of the exploitation of this work for future publications, a few interesting questions could enhance the current study:

- Has the day 194 been compared to any other time point, such as the baseline ? Is there any changes observed in any population? It would indeed be interesting to understand whether the characteristics of the memory T cells here, results from a previous infection with hookworms ( other other endemic pathogen) or whether it results from the vaccination. CTLA-4 might be high regardless of the vaccination with Na proteins. What happens in the HBV control group.

- Do the results confirm the results from figure 2, is the sensitivity/resolution any better in particular for the type 2 cytokines. One might wonder whether stimulating with APR-1 as well might be interesting.

As total PBMC have been stimulated, are T cells the only or main TNF/IL-2 producing cells ? Is there other population responding to the restimulation.

Figure 4 :The authors speculate in the discussion that the cytokine production in response to APR-1 stimulation could be due to a too low level of production, because of insufficient priming with NA-APR-1. Is CTLA-4 blockade enhancing cytokine production in response to APR-1 stimulation as well ? If yes, which cytokines are increased ?

**Conclusions**

-Are the conclusions supported by the data presented?

-Are the limitations of analysis clearly described?

-Do the authors discuss how these data can be helpful to advance our understanding of the topic under study?

-Is public health relevance addressed?

Reviewer #1: Yes, conclusions supported by data presented

Reviewer #2: The conclusion is well written and support the data presented. Just a comment for the authors: 

The authors state “No significant changes over time were detected for IFN-γ, TH2cytokines (IL-4/-5/-13) or IL-10 in response to Na-GST-1” As type 2 responses are usually associated with protection against helminths, these results are quite surprising, could the authors expand further their comments on this in the discussion. In particular on what could be expected in terms of efficacy of the treatment. To my knowledge it is unknown even in animal models how the immunisation with APR-1 is working. Are only the antibodies required or do the cellular immune response need to be polarised as well. 

The subclass of IgG in animal model for APR-1 have been proposed to be IgG1 and IgG4, which depends on IL-4 and IL-10 respectively for class switching. A poor modified type 2 response could explain the low antibody response to APR-1. As immunization with GST-1 has been shown to give rise to IgG3 and IgG1, the type 2 cytokines might not be as required ? 

overall, it might be that it is not that much the dose of APR-1 that needs to be increased but maybe that an other adjuvant is required.

**Editorial and Data Presentation Modifications?**

Reviewer #1: Abstract/Methods: It would be useful to provide more information on methods in the abstract: x (32 or 24?) Gabonese volunteers were randomized to vaccination with three doses of a low-dose (n=8) or high-dose (n=10) combination Na-GST-1 or Na-APR-1 or Hepatitis B vaccine (n=6) over a 6-month period. Blood was collected pre-vaccination on days 0, 28, and 180 and 2-weeks after each vaccine dose on days 14, 42, and 194. PBMCs were isolated from blood samples…etc.

Abstract/Results: would be good to include some relevant data?

Introduction: Lines 114-115: Please correct: “(Ascaris lubricoides, Trichuris trichura, Necator ankylostoma and Strongiloides stercoralis)”

Reviewer #2: (No Response)

**Summary and General Comments**

Reviewer #1: A short report of the cytokine response to two hookworm vaccine candidates. The paper requires a bit more information in the abstract and method but otherwise is clear and appears scientifically sound.

Reviewer #2: Mouwenda et al, describe the specific CD4 T cell response after vaccination with the two-vaccine candidates Na-GST1 and Na-APR-1 in an endemic population. The authors show limited specific response of CD4 T cells to Na-APR1, while specific T cells to Na-GST-1 were mostly found to release IL-2 and TNFa. Using mass cytometry the authors further show that TNFa producing memory CD4 T cells express high level of the inhibitory molecule CTLA-4. In vitro blockade of CTLA-4 increase TNFa production upon specific Na-GST-1 stimulation, suggesting that an intervention with anti-CTLA-4 in vaccinated human could be beneficiary.

PLOS authors have the option to publish the peer review history of their article (what does this mean?). If published, this will include your full peer review and any attached files.

Reviewer #1: No

Reviewer #2: No

Figure Files:

Data Requirements:

Reproducibility:

References

---

## [Editor Report · Decision Letter 1]

14 Aug 2021

Dear Mrs Mouwenda,

We are pleased to inform you that your manuscript 'Characterization of T cell responses to co-administered hookworm vaccine candidates Na-GST-1 and Na-APR-1 in healthy adults in Gabon' has been provisionally accepted for publication in PLOS Neglected Tropical Diseases.

Best regards,

Keke C Fairfax, PhD

Deputy Editor

Keke Fairfax

Deputy Editor

---

## [Editor Report · Acceptance letter]

14 Sep 2021

Dear Mrs Mouwenda,

We are delighted to inform you that your manuscript, "Characterization of T cell responses to co-administered hookworm vaccine candidates Na-GST-1 and Na-APR-1 in healthy adults in Gabon," has been formally accepted for publication in PLOS Neglected Tropical Diseases.

Best regards,

Shaden Kamhawi

co-Editor-in-Chief

Paul Brindley

co-Editor-in-Chief
